# The Nature of the Enthalpy–Entropy Compensation and “Exotic” Arrhenius Parameters in the Denaturation Kinetics of Proteins

**DOI:** 10.3390/ijms241310630

**Published:** 2023-06-25

**Authors:** Alexey V. Baklanov, Vitaly G. Kiselev

**Affiliations:** 1Institute of Chemical Kinetics and Combustion SB RAS, 3 Institutskaya Street, 630090 Novosibirsk, Russia; vitaly.kiselev@gmail.com; 2Department of Physics, Novosibirsk State University, 1 Pirogova Street, 630090 Novosibirsk, Russia

**Keywords:** protein, unfolding, polyglycine dimer, Arrhenius parameters, enthalpy–entropy compensation, transition state theory, “completely loose” transition state, solvation effect

## Abstract

Protein unfolding is a ubiquitous process responsible for the loss of protein functionality (denaturation), which, in turn, can be accompanied by the death of cells and organisms. The nature of enthalpy–entropy compensation (EEC) in the kinetics of protein unfolding is a subject of debate. In order to investigate the nature of EEC, the “completely loose” transition state (TS) model has been applied to calculate the Arrhenius parameters for the unfolding of polyglycine dimers as a model process. The calculated Arrhenius parameters increase with increasing dimer length and demonstrate enthalpy–entropy compensation. It is shown that EEC results from the linear correlations of enthalpy and entropy of activation with dimer length, which are derived directly from the properties of the transition state. It is shown that EEC in solvated (hydrated, etc.) proteins is a direct consequence of EEC in proteins themselves. The suggested model allows us also to reproduce and explain “exotic” very high values of the pre-exponential factor measured for the proteins unfolding, which are drastically higher than those known for unimolecular reactions of organic molecules. A similar approach can be applied to analyzing the nature of EEC phenomena observed in other areas of chemistry.

## 1. Introduction

Proteins are the key life agents in nature. They possess unique functionality in the native states operating in the cells of all known organisms. These native states are assembled by the folding of the protein chain into a secondary, tertiary, or even quaternary structure. The unfolding of these structures, defined as denaturation, results in the loss of their functionality, which can be followed by the death of the cells and organisms. Temperature is the most widely present factor of denaturation. Thus, the temperature dependence of the denaturation kinetics is of utmost interest in biology and chemistry. Thermally induced denaturation of proteins often proceeds as a single-exponential process (two-state kinetics, where the native state is a reagent and the denatured state is a product) with the temperature dependence of the rate constant empirically described with the Arrhenius equation.
(1)$$k=A\times e^{-\frac{E_{a}}{RT}}.$$
or, more specifically, with the transition state theory (TST) equation:(2)$$k=\frac{k_{B}T}{h}\cdot e^{\frac{∆S^{\neq }}{R}}\cdot e^{-\frac{∆H^{\neq }}{RT}},$$
where $∆S^{\neq }$ and $∆H^{\neq }$ are the entropy and enthalpy of activation [1,2]. Rosenberg et al. noted the similarity in the thermal kinetic behavior of proteins and many other microorganisms, viz., yeasts, viruses, and bacteria [3]. They revealed that $∆S^{\neq }$ and $∆H^{\neq }$ for the thermal inactivation of these microorganisms obeys the enthalpy–entropy compensation (EEC) equation:(3)$$∆S^{\neq }=a\cdot ∆H^{\neq }+b,$$
with parameters *a* and *b* being similar to those for protein denaturation [3]. This allowed Rosenberg et al. to infer the thermal denaturation of proteins to be a cause of the thermal death of microorganisms. To the present time, much more data on the kinetics of the denaturation of proteins and thermal deactivation of different microorganisms have been collected. They also follow the EEC equation with similar parameters [4,5,6,7,8,9,10]. Thus, the kinetic data obtained for individual proteins and those for the thermal inactivation of microorganisms can be considered in a unified way. Note that the values of Arrhenius parameters are very widely scattered. Qin et al. summarized the experimental literature data presented in the review by He and Bischof [7] and reported that the activation energy *E_a_* in various protein systems varies between about 25 and 200 kcal/mol and the pre-exponential factor *A*, between 10^9^ and 10^129^ s^−1^ ([9], Figure 1). Even higher numbers for *A* and *E_a_* (up to 10^218^ s^−1^ and about 310 kcal/mol, respectively) are reported in the data reviewed by Wright ([8], Tables 1,3,4). It is worth emphasizing that the pre-exponential factors for these processes are often drastically higher than their counterparts for the unimolecular reactions of small organic molecules. The latter usually have the *A* factors at 10^13^–10^14^ s^−1^. A large amount of data for relatively small organic molecules are collected in the well-known monography of Holbrook and Robinson [11]. Within the concepts of transition state theory, the authors of ref. [11] interpreted the pre-exponential factors higher than ~10^15^ s^−1^ as characteristic of the reactions with “loose” transition states. However, even the highest *A* values around 10^18^ s^−1^ given in ref. [11] are far below the upper bounds for proteins, which reach values greater than 200 orders of magnitude. These experimentally measured ultrahigh values are “exotic” and should be explained.

The enthalpy–entropy compensation, also known by other names (e.g., the kinetic compensation effect, the isokinetic and isoequilibrium relations), is a very ubiquitous phenomenon occurring in various fields of science, including chemistry, physics, materials science, biology, and geology, with various concepts proposed to date [12,13,14,15,16,17,18,19,20,21,22,23,24,25,26,27,28,29,30,31,32,33,34,35]. Liu and Guo [24] and Freed [30] listed a very wide range of specific areas where this compensation phenomenon is observed, which include Langmuir adsorption, protein and nucleic acids chemistry, depolymerization, dissociation of lipid complexes, drug–receptor binding, thermal isomerization and substitution reactions, and many other processes. A comprehensive review of the concepts proposed to explain EEC and their evolution is given by Liu and Guo [24]. Several concepts have been elaborated to explain the EEC observed for the enthalpy and entropy of proteins unfolding, which characterize the thermodynamics of the equilibrium. Since the work by Lumry and Rajender [12], many authors have hypothesized that the physical origin of EEC has its basis not in the unfolding of proteins themselves but in the compensation taking place in the reorganization of solvent [16,30] or specifically in the reorganization of water [15,17,23,24,26,36], which accompany the unfolding of proteins in a solution. We should also mention the works of Gilli et al. [15] and Dunitz [17], who considered this effect to be an intrinsic property of hydrogen bonds. In analyzing the thermal stability of collagen as a function of its hydration, Miles and Ghelashvili inferred the linear relationship between entropy and enthalpy using the concept of entropy of a “polymer-in-a-tube” [21]. Only one of the suggested concepts explains EEC in the Arrhenius parameters of the rate constant of protein denaturation. This interpretation is a statistical one, which considers EEC behavior as a data-handling artifact that is not provided by chemical causation [13,31]. According to this interpretation, the scatter of the experimental data provided by the measurement errors gives rise to the linear distribution of highly correlated estimates of enthalpy and entropy of activation.

In this work, we show that EEC is the intrinsic property of the denaturation kinetics of the proteins themselves. In order to analyze the EEC behavior at a molecular level, we suggest the “completely loose” transition state model, which allows us to calculate the Arrhenius parameters for proteins unfolding using transition state theory (*TST*). The unfolding is modeled with the dissociation of the polypeptide dimer into two single polypeptide chains. In order to consider the EEC effect without any irrelevant complications, we have considered polyglycine as the simplest case of polypeptide. Figure 1 shows the structure of the β-sheet dimer of polyglycine formed by the β-strand polypeptides. Each of the two polypeptide chains contains the glycine fragments –NH-CH_2_-C(O)- with the terminal groups CH_3_CO- and CH_3_NH-. Two chains of polypeptides are bound with interchain hydrogen bonds. The hydrogen bonds determine the structure of the folded states of proteins [2,37]. In Figure 1, each link contains two interchain hydrogen bonds. The calculations were carried out for the dimers containing 1, 2, 3, 4, 6, 8, 12, and 16 links. These dimers are further referred to as Dim1-Dim16, and the corresponding monomers as Mon1-Mon16. As an instructive example, the dimer structure shown in Figure 1 comprises 4 links (Dim4).

We model the unfolding process by unimolecular dissociation of the dimer into two single monomer molecules of polyglycine.
(4)$$Dimer\;\overset{k}{\to }\;2Monomer$$

The calculations of the rate constant *k* were carried out within TST theory with a “completely loose” transition state model described below. The calculations of the dissociation rate constants were performed for the dimers of the glycine polypeptide of different lengths.

## 2. Results

### 2.1. “Completely Loose” Transitionj State Theory Model

The transition state theory (*TST*) equation for the rate constant of a unimolecular reaction reads as:(5)$$k_{TST}=\frac{\kappa T}{h}\cdot \frac{Q_{}^{\neq }}{Q_{Dim}}\cdot e^{-\frac{E_{0}}{\kappa T}}.$$

Here, $E_{0}$ is the energetic barrier and $Q^{\neq }$ with $Q_{Dim}$ are the partition functions of the transition state and initial dimer, respectively. For the sake of brevity, the electronic state’s degeneracy and symmetry factors are omitted. There is no well-defined energy maximum on the reaction coordinate of homolytic bond dissociation, but there is a maximum for the Gibbs free energy [38]. Given the current state of quantum chemical calculations regarding system size, level of theory, and sampling approach [39], calculating the full free energy profile along the dissociation reaction coordinate for a polypeptide dimer containing hundreds of atoms seems to be unrealistic at present. We observed very high pre-exponential factors for proteins unfolding points to the need to use the model of the transition state of extreme looseness. The potential curve corresponding to the hydrogen bond breaking, as well as the van der Waals interactions, monotonically rises to the dissociation limit [40]. Therefore, the value of the energy barrier for the dimer dissociation $E_{0}$ can be taken equal to the enthalpy of the dissociation process (4) at *T* = 0 K ($∆H_{0}^{0}$). To calculate the TS partition function $Q^{\neq }$ we apply the “completely loose” transition state (*TS*) approach. According to the definition of Quack and Troe, the “completely loose” *TS* corresponds to the two independent product moieties not restricted in their dynamics apart from the fixed reaction coordinate [38]. In our case, the products are polypeptide monomers. We consider the distance *R* between the centers of mass of the monomer units to be a reaction coordinate. The variational *TS* is proposed to be a pair of freely rotated Mon molecules at a fixed *R*-value ($R^{\neq }$) as shown in Figure 2.

We derive the partition function of *TS* from that of products in a way:(6)$$Q_{}^{\neq }=Q_{c}^{\neq }\cdot Q_{tr}^{\neq }.$$

Here, $Q_{c}^{\neq }$ is the partition function of the group of the low-amplitude conserved modes, which are similar to those of products and $Q_{tr}^{\neq }$—of the group of large amplitude transitional/external-rotational modes. This approach is similar to that used by Robertson et al. in their flexible transition state approach [41], but they derived the partition function of conserved modes ($Q_{c}^{\neq }$) from the modes of the reagent of unimolecular reaction. In our case, the low-amplitude conserved modes include vibrations, which correspond to those for free Mon molecules with a partition function equal to $Q_{c}^{\neq }=Q_{vib,Mon}{}^{2}$, where $Q_{vib,Mon}$ is the vibrational partition function of the Mon molecule. We neglect the interaction between the Mon molecules in TS. This allows us to consider them as rigid free rotors located at a fixed distance $R^{\neq }$ between their centers of mass. The $Q_{tr}^{\neq }$ group of the transitional modes for the “completely loose” TS at the fixed *R* can be represented by the equation:(7)$$Q_{tr}^{\neq }=Q_{pd}\cdot Q_{fr,1}^{}\cdot Q_{fr,2}^{},$$
where $Q_{fr,1}^{}$ and $Q_{fr,2}^{}$ are the partition functions for the free rotation of monomers ($Q_{rot,Mon}$), and $Q_{pd}=\frac{2\cdot \mu \cdot {\left(R^{\neq }\right)}^{2}\cdot \kappa T}{\hbar^{2}}$ is the rotational partition function of the pseudo-diatomic species formed from two point masses of monomer units located at a distance $R^{\neq }$. Equation (7) can be derived from the result of the paper by Robertson et al. [41], Equation (19) with a hindering function taken to be G (R, T) = 1, which is correct for the case of non-interacting Mon units. Finally, the Equation (5) can be rewritten as
(8)$$k_{TST}=\frac{\kappa T}{h}\cdot \frac{Q_{pd}\cdot Q_{rot,Mon}^{2}\cdot Q_{vibr,Mon}^{2}}{Q_{rot,Dim}\cdot Q_{vibr,Dim}}\cdot e^{-\frac{∆H_{0}^{0}}{\kappa T}}$$

In *TS*, the distance between the centers of mass of monomers is extended as compared with that in the initial dimer (≈5 Å) by an extra 2 Å yielding $R^{\neq }$ = 7 Å. This extension exceeds the value of 0.5 Å, supposed by Sheu et al. [40] to be sufficient for breaking interchain hydrogen bonds. A somewhat arbitrary choice of $R^{\neq }$ value is unimportant for the results obtained because only $Q_{pd}$ contains $R^{\neq }$ value in Equation (8). Thus, the effect of its variation is small and equal for all $k_{TST}$ values calculated using Equation (8).

Equation (8) allows us to obtain the equation for Arrhenius activation energy.
(9)$$E_{a}=RT^{2}\cdot \frac{dlnk_{TST}}{dT}=\frac{7}{2}RT+∆H_{0}^{0}+2\cdot {\langle E\rangle }_{vib,Mon}-{\langle E\rangle }_{vib,Dim},$$
where ${\langle E\rangle }_{vib,Mon}$
*and*
${\langle E\rangle }_{vib,Dim}$ are average vibrational energy at a temperature of *T* for monomer and dimer molecules, respectively. Pre-exponential factor *A* is then derived from calculated $k_{TST}$ and $E_{a}$ values. The energy, structure, and vibrational wavenumbers have been calculated within the DFT approach discussed below. Partition functions and average thermal energy were calculated within the harmonic oscillator-rigid rotor approximation.

### 2.2. Polyglycine Monomer and Dimer Structure

The calculated structures of Mon4 and Dim4, as examples, are shown in Figure 3. The monomer geometry corresponds to planar all-trans conformation. It is worth mentioning that the calculated energy of this conformation is slightly higher than that for the helical conformation of the monomers of any length. However, the calculated free energy at 298 K of all-trans structure is lower than that of its helical counterpart for all considered MonN. Thus, we considered henceforth only the all-trans conformation of the monomers Mon1-Mon16. The optimized Dim4 geometry is similar to the antiparallel chain-rippled sheet structure of crystalline polyglycine I described by Moore and Crimm [42]. All Dim1-Dim16 geometries belong to the *C_i_* point group.

### 2.3. The Enthalpy–Entropy Compensation

The Arrhenius parameters calculated using Equations (8) and (9) are shown in Figure 4a. The entropy and enthalpy of activation, presented in Figure 4b, were calculated according to Equations (A1) and (A2) in Appendix A. As seen in Figure 4, the calculated Arrhenius parameters vary within very broad intervals, e.g., the activation energy (*E_a_*) values lie between 20 and 140 kcal/mol, and the pre-exponential factors (*A*) between 10^21^ and 10^105^ s^−1^. Figure 4 demonstrates the very consistent EEC effect in a row of dimers of varied lengths.

Found EEC behavior is a consequence of the linear correlations of both enthalpy and entropy of activation with the length (number of links) of the dimer, which is shown in Figure 5. 

## 3. Discussion

### 3.1. The Enthalpy–Entropy Compensation

Very precise Enthalpy-Entropy compensation demonstrated in Figure 4 is a consequence of the linear correlations of both enthalpy and entropy of activation with the length (number of links) of the dimer, which is shown in Figure 5. In Appendix A, the linear correlations $∆H^{\neq },∆S^{\neq }-Number\;of\;Links\;\left(N\right)$ and their increments are directly derived from the properties of the transition state of polypeptide unfolding. These linear correlations are determined by similarities in the structure and vibrational characteristics of the interchain hydrogen bonding frames corresponding to the chain links. The quantitative parameters obtained with our model calculations fit well the values expected for unfolding of proteins. 

The activation enthalpy rises with an increment of 7.9 kcal/mol per link of the dimer chain (see Figure 5), which corresponds to ≈4 kcal/mol per one interchain hydrogen bond. This number agrees well with the value of 4.2 kcal/mol of N−H⋯O=C hydrogen bond energy in non-polar solvent found by Klotz [43] for a prototype functional group.

The slope of the straight line in Figure 4b corresponds to 1000·*a*, where *a* is a coefficient from EEC Equation (3). This slope allows us to determine the temperature of so-called complete compensation T_c_ = 1/*a*, where the exact compensation of the increase in activation enthalpy $∆H^{\neq }$ by the entropy contribution $T\cdot ∆S^{\neq }$ takes place. The value *T_c_* = 302.5 K, determined from the slope in Figure 4b, is very close to *T_c_* values extracted from massive experimental data for protein denaturation in refs. [3,4,5,6,7,8,9,10]. Values of *T_c_* obtained by Rosenberg et al. (325–331 K) [3] or numbers we extracted from the parameters of ln*A* − E_a_ dependences obtained by He and Bischof (315 K) [7], Wright (314 K) [8], Qin et al. (314–331 K) [9], and Yap et al. (305 K) [10], differ from the value, obtained with our model calculations, by no more than 10%. This confirms the adequacy of the suggested model of protein unfolding.

### 3.2. “Exotic” Values of Pre-Exponential Factor

The activation entropy increment of 26.0 e.u. per link corresponds to the increase of pre-exponential factor by 5·10^5^ times, yielding the *A* values within the interval 10^20^–10^105^ s^−1^ for the dimers D1–D16. These high *A* numbers are due to the “completely loose” TS properties. The interchain hydrogen bonds give some rigidity to the structure of the dimer. The breaking of these bonds makes the structure very loose and results in the decrease of the frequencies of vibrations localized within a link of the dimer. This change leads to a dramatic increase in the entropy of the TS structure compared with the dimer and, consequently, to high pre-exponential factor values. These arguments are qualitatively similar to those which are used in ref. [11] to explain the “high” pre-exponential factor values at a level of 10^15^–10^18^ s^−1^ for the unimolecular reactions of organic molecules. But this similarity is only qualitative, not quantitative. The key feature of the dissociation of the considered polypeptide dimers is that a much higher number of the degrees of freedom become loose in *TS* compared to the more rigid initial dimer. This argument is illustrated by the fact that two non-interacting monomers Mon16 in *TS*, have 174 vibrations with the wavenumbers lower than 100 cm^−1^, which is notably higher than 127 vibrations with these low wavenumbers in dimer Dim16. This big increase in the number of low-frequency vibrations in the *TS* compared with the initial dimer is the key factor that provides a dramatic increase in the activation entropy and, therefore, leads to “exotic” very high values of A factors. The same arguments are applicable to the explanation of similar very high numbers observed experimentally for the unfolding of proteins discussed in the Introduction.

### 3.3. Extrapolation of the Results of the Unfolding of Polypeptides and Proteins with Other Amino Acid Composition

The results of our calculations show that the values of the Arrhenius parameters of the unfolding rate constant correlate with the number of breaking interchain hydrogen bonds. The higher the number of the bonds to be broken, the higher both the activation energy and pre-exponential factor of the unfolding rate constant. Our conclusions are obtained in the particular case of polyglycine dimers dissociation. However, the similar results can be anticipated for the unfolding of proteins with other amino acid compositions. The interchain hydrogen bonds N−H⋯O=C are expected to have close energy values for any amino acid composition of the proteins to be unfolded [43]. This allows us to expect the linear correlation of the activation enthalpy of the unfolding process with the number of breaking interchain hydrogen bonds for these proteins as well. The most essential change in the vibrational wavenumbers in TS compared with the initial dimer occurs for the vibrations of the frame containing interchain hydrogen bonds. The structures of these frames are similar for any pairs of amino acids. Here we can mention that the distance *N*⋯O in the interchain hydrogen bonds N−H⋯O=C in the dimer of polyglycine is equal to 2.9 Å [42], which corresponds to the values 2.8–3.0 Å [40] for different pairs of amino acid residues in the other proteins. This, in turn, results in the linear correlation for the entropy of activation in the unfolding process involving the breaking of the interchain hydrogen bonds between the different pairs of amino acids. The manifold of the pairs of amino acid residues interconnected with interchain hydrogen bonds can be considered the discrete structural units making stable a secondary or a higher order structure of the native state of a protein. Breaking of the interchain H-bonds in each unit results in similar incremental contributions to activation enthalpy and entropy, ultimately rendering the EEC behavior for proteins unfolding. This similarity is also confirmed by the close values of the temperature of complete compensation T_c_ for unfolding polyglycine dimer and proteins studied experimentally (see Section 3.2). Our results and their extrapolation are relevant to the gas-phase unfolding process and can be applied to dry proteins as well.

### 3.4. Extrapolation of the Results of the Unfolding of Polypeptides and Proteins in Water and Other Solvents

The “softening” of hydrogen bonds takes place in a water solution. The influence of water is due to the interaction with H_2_O molecules located near the atoms participating in the formation of interchain hydrogen bonds [44]. The hydrated folded and unfolded proteins can also be considered as a chain of links comprised of amino acid residues with coordinated water molecules. Similar to the results of our calculations, the linear correlations of both enthalpy and entropy of activation with the number of breaking hydrogen bonds remain valid. Thus, the EEC behavior is expected to be valid for hydrated proteins as well. The same conclusion is also true for the “exotic” values of the Arrhenius parameters. However, the quantitative parameters of the correlations between the enthalpy (entropy) of activation and the number of breaking interchain hydrogen bonds are different. The maximal value of Arrhenius activation energy *E*_a_ (≈138 kcal/mol) obtained in our calculations correspond to the breaking of 32 interchain hydrogen bonds in a single gas-phase dimer. In a water solution, the similar values of Arrhenius parameters should correspond to the breaking of a higher number of hydrogen bonds. More specifically, Shea et al. performed the molecular dynamics simulation of the of the H-bonding in β-hairpin in vacuum and water [40]. They found that the breaking of the hydrogen bond in a β-sheet in water has an activation energy of about 1.5 kcal/mol, which is lower by about 3 kcal/mol as compared with the case of an isolated β-sheet. This effect of water is found to be in agreement with the experimental observations [40]. This means that the unfolding in water with the upper indicated Arrhenius activation energy *E*_a_ (≈138 kcal/mol) corresponds to a breaking of about 100 interchain hydrogen bonds.

Here we can also show that the EEC for proteins unfolding in a solvent (water, etc.) can be directly inferred from EEC taking place for the unfolding of isolated proteins (gas-phase unfolding) with the thermodynamic arguments. We will consider the effect of solvents in a way similar to that used by Liu et al. [23]. The authors represented the total enthalpy and entropy of protein unfolding in water ($∆H_{un,w},\;∆S_{un,w}$) as the sums of the terms relevant to the unfolding of the protein itself ($∆H_{un}$, $∆S_{un}$) and those relevant to the reorganization of water during protein unfolding ($∆H^{r},∆S^{r}$). We can use similar equations for the enthalpy and entropy of activation.
(10)$$∆H_{un,w}^{\neq }=∆H_{un}^{\neq }+∆H_{}^{\neq ,r}$$
(11)$$∆S_{un,w}^{\neq }=∆S_{un}^{\neq }+∆S_{}^{\neq ,r}.$$

According to the general thermodynamic relations, the enthalpy of water reorganization (or any other solvent) during a chemical reaction is exactly compensated by the corresponding change of entropy ($∆H^{r}=T\cdot ∆S^{r}$) [36]. Finally, the compensation takes place if the corresponding process is in the equilibrium. One of the basic assumptions used in transition state theory is the assumption of the equilibrium between the reagents and the transition state. This allows us to assume the same relation between the enthalpy and entropy of activation $∆H^{\neq ,r}=T\cdot ∆S^{\neq ,r}$. With the use of the equation for unfolding the protein itself $∆S_{un}^{\neq }=a\cdot ∆H_{un}^{\neq }+b=\frac{∆H_{un}^{\neq }}{T_{c}}+b$, we finally get the equation $∆S_{un,w}^{\neq }=\frac{∆H_{un}^{\neq }}{T_{c}}+\frac{∆H^{\neq ,r}}{T}+b\;$, which at *T* ≈ *T_c_* reads as:(12)$$∆S_{un,w}^{\neq }\approx \frac{∆H_{un,w}^{\neq }}{T_{c}}+b=a\cdot ∆H_{un,w}^{\neq }+b$$

Therefore, if the measurements of Arrhenius parameters are carried out at a temperature close to the temperature of complete compensation *T_c_* for the unfolding of proteins themselves, the activation entropy and enthalpy for unfolding of solvated proteins, though different in magnitude, should lie on the same $∆S^{\neq }-∆H^{\neq }$ linear dependence. This conclusion is confirmed by our model calculations, which show that the temperature *T_c_* for the unfolding of polyglycine dimers themselves is very similar to the values of *T_c_* obtained experimentally for the unfolding of proteins, many of which are studied in water.

## 4. Materials and Methods

### 4.1. Quantum-Chemical Calculations

The geometries of all structures corresponding to the stationary points on the potential energy surface (PES) of the species studied were fully optimized using density functional theory at the B3LYP/6-31G(d) [45] level of theory. The numerical integration of the exchange-correlation terms of the density functionals was carried out using superfine grids. All calculations were performed using the Gaussian 16 suite of programs [46]. Zero-point and average thermal energies were computed at the same DFT level.

### 4.2. Transition State Theory Calculations

The model of “Completely Loose” Transition State has been suggested to calculate Arrhenius parameters of the rate constant of polyglycine dimer unfolding. This "completely loose" TC corresponds to two independent product molecules free in their dynamics, except for a fixed reaction coordinate, which is the distance between the centers of mass of the product molecules. 

## 5. Conclusions

The dissociation process of polyglycine dimers of varied lengths has been investigated as a model of protein unfolding. The “completely loose” transition state (TS) model is suggested, which allows us to calculate the kinetic parameters for these processes within transition state theory. This “completely loose” TS presents two non-interacting monomer chains at a fixed distance between the centers of mass of these chains. The results of TST calculations demonstrate the enthalpy–entropy compensation (EEC) for the Arrhenius parameters of the unfolding of dimers of varied lengths. EEC is a result of the linear correlations of enthalpy and entropy of activation with dimer length, which are derived directly from the obtained equation for the rate constant of unfolding. The similarity in energy values and structure for the frames provided by interchain hydrogen bonds N−H⋯O=C between different amino acid residues allows us to expect a similar correlation for activation enthalpy and entropy in the unfolding of proteins. The EEC behavior for the unfolding of proteins themselves and known compensation between the enthalpy and entropy of solvent reorganization in any equilibrium chemical process allow us to predict directly the EEC behavior for proteins unfolding in water or other solution. Moreover, the data for the unfolding of non-solvated and solvated proteins are predicted to lie on the same $∆S^{\neq }-∆H^{\neq }$ linear dependence. This conclusion is confirmed by the fact that the calculated temperature of complete compensation T_c_ for EEC in a free dimer of polypeptide coincides with experimentally measured values of T_c_ for protein denaturation in water with an accuracy better than 10%. The results obtained indicate that the EEC behavior is the intrinsic property of the denaturation process in the dry or hydrated forms of proteins.

The suggested TS model allows us also to reproduce and explain “exotic” very high numbers for the pre-exponential factor *A* values measured for the proteins unfolding, which are drastically higher than those known for unimolecular reactions of organic molecules. The highest *A* value we obtained for polypeptide dimers containing 32 interchain hydrogen bonds is 10^105^ s^−1,^ and even higher values are predicted for the unfolding of longer polypeptide dimers. The key factor providing these very high *A* values is a big increase in the number of low-frequency vibrations in TS compared to the initial folded protein.

The power of the suggested TS model in the explanation of EEC and “exotic” values of Arrhenius parameters allows us to hope for its productive application in protein science. We also hope that a similar TST approach can be applied to analyze the nature of EEC phenomena observed in many other areas of chemistry.

## Figures and Tables

**Figure 1 ijms-24-10630-f001:**
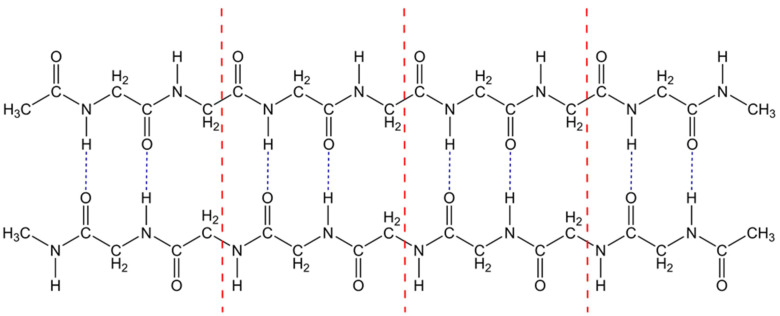
Structure of the polyglycine dimer. The β-sheet polyglycine dimer is formed by β-strands of polypeptides with four links (Dim4), each containing two interchain hydrogen bonds. Red dash-dotted lines separate the adjoining links. Blue short-dashed lines indicate interchain hydrogen bonds.

**Figure 2 ijms-24-10630-f002:**
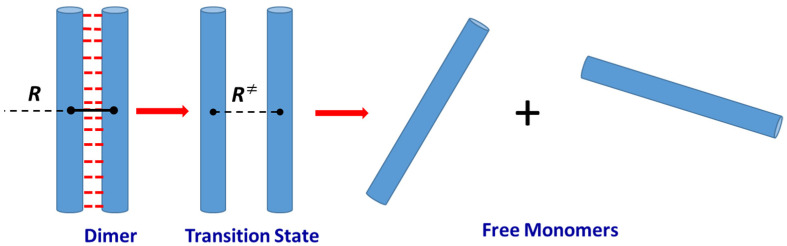
Scheme of polypeptide dimer dissociation within a “completely loose” transition state approach.

**Figure 3 ijms-24-10630-f003:**
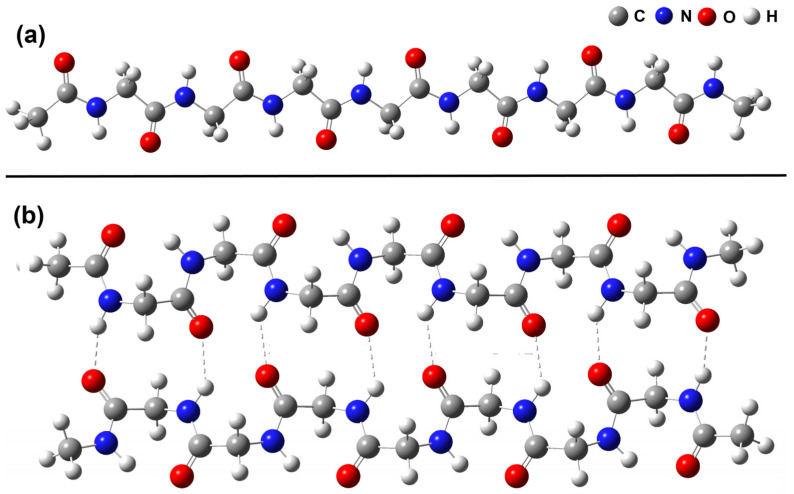
B3LYP/6-31G(d) optimized structure of polyglycyne: (**a**)—monomer Mon4 and (**b**)—dimer Dim4. The atomic symbols: C—grey, O—red, N—blue, and H—white. The dashed lines indicate interchain hydrogen bonds.

**Figure 4 ijms-24-10630-f004:**
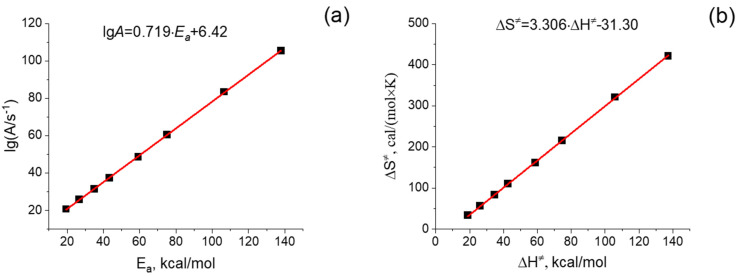
(**a**) The linear fit of lg*A*−*E_a_* correlation and (**b**) the linear fit of $∆S^{\neq }-∆H^{\neq }$ correlation. The preexponential factor *A*, activation energy *E*a, entropy $∆S^{\neq }$ and enthalpy $∆H^{\neq }$ of activation correspond to the rate constants of the dissociation of the polyglycine dimers of different lengths (DimN with N = 1, 2, 3, 4, 6, 8, 12, 16). All presented values are calculated at T = 298.15 K.

**Figure 5 ijms-24-10630-f005:**
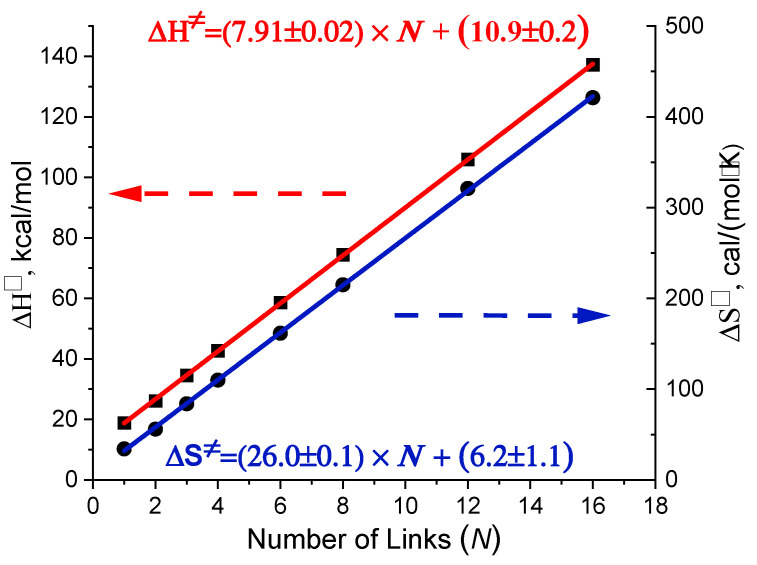
Calculated enthalpy and entropy of activation as a function of the number of links (N) in dimers. Parameters of the linear fits of the presented data are given in the color of corresponding line.

## Data Availability

Data are contained within the article.

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
