# Peer review of "The Nature of the Enthalpy–Entropy Compensation and “Exotic” Arrhenius Parameters in the Denaturation Kinetics of Proteins"

_ijms, 2023, doi:10.3390/ijms241310630_

Round 1

Reviewer 1 Report

The authors investigated the nature of Enthalpy-Entropy compensation (EEC) in the kinetics of protein unfolding. On this purpose, the "completely loose" Transition State model has been applied to calculate the Arrhenius parameters for the unfolding of polyglicine dimers, taken as a prototype. 

The main results are the calculation of the Arrhenius parameters by EEC correlation, that permits to predict the EEC behavior for proteins unfolding in water or other solutions. A linear dependence between ∆S and −∆H was found; this is confirmed by the fact that the calculated temperature of complete compensation Tc for EEC in a free dimer of polypeptide coincides with experimentally measured values of Tc for proteins denaturation in water with accuracy better than 10%. The authors conclude that the EEC behavior is an intrinsic property of the denaturation process in hydrated and non-hydrated forms of proteins. 

The authors apply their suggested TS model to reproduce and explain “exotic” numbers for the preexponential factor A values measured for the proteins unfolding (e.g. A=10105 s-1) . This extremely high value is explained by  a big increase in the number of low-frequency vibrations in TS as compared with initial folded protein.

I have a minor comment and a major comment that the authors should consider.

Minor comment (lines 129-131): The authors state: "Unfortunately, in the case of polypeptide dimers, the calculation of the full free energy profile along the dissociation reaction coordinate is currently non- solvable task". The authors should motivate why it is an unsolvable task and possibly provide a reference.

Major comment: I am concerned about the "exotic" value A=10105. Did the authors provide an explanation about this phenomenon?

However, I wonder if this value is based in fact. Are there any experimental evidences of such a high value? Or it has simply been inferred from log (A) vs. Ea correlation in Fig. 4? The authors reported a reference [11], but I could not find any reaction with such a high values about the pre-exponential factor. The authors should report some examples and more detailed references (page, table, figures, etc.). Are there any specific references to protein denaturation?

Just in case there are no experimental evidence for this value, I would recommend not publishing this result. In this case, the manuscript should be evaluated again after the revision process. 

Reviewer 2 Report

This manuscript describes a computational study exploring the Enthalpy-Entropy compensation model for protein unfolding using a poly-glycine model system. Their model uses a transition-sate model where the dimer is in a closely associated state, that they refer to as “completely-loose” calculate Arrhenius parameters for the unfolding process. These parameters increase with dimer length, which the authors conclude is due to Enthalpy-Entropy compensation. The authors suggest that this model explains why protein unfolding traditionally have very high preexponential factors compared to organic molecules.  

Overall, this is a very clearly written paper. It is the first paper I can remember reviewing recently where I did not find a single typo. While I am not a computational chemist, the study was clearly described and the logic for various equations was presented in a way that was easy to interpret.

My one major critique is that this is a very simple model system that make sense computationally due to the number of atoms. However, the authors did not make it very clear how this would correlate with more complex systems.  Even a GAGAGAGA peptide would more closely resemble a real protein. Since the authors are trying to make the case that this resembles the unfolding process in a real protein, a clearer description of why this is a valid model system would make this a stronger manuscript.

Round 2

Reviewer 1 Report

The authors satisfactorily replied to the referee’s comments. I recommend the paper for publication.